

# Deconstructing little strings with $\mathcal{N}=1$ gauge theories on ellipsoids

Joseph Hayling⋆, Rodolfo Panerai† and Constantinos Papageorgakis‡

CRST and School of Physics and Astronomy
Queen Mary University of London
Mile End Road, London E1 4NS, UK

⋆ j.a.hayling@qmul.ac.uk †r.panerai@qmul.ac.uk ‡c.papageorgakis@qmul.ac.uk

## Abstract

A formula was recently proposed for the perturbative partition function of certain $\mathcal{N}=1$ gauge theories on the round four-sphere, using an analytic-continuation argument in the number of dimensions. These partition functions are not currently accessible via the usual supersymmetric-localisation technique. We provide a natural refinement of this result to the case of the ellipsoid. We then use it to write down the perturbative partition function of an $\mathcal{N}=1$ toroidal-quiver theory (a double orbifold of $\mathcal{N}=4$ super Yang–Mills) and show that, in the deconstruction limit, it reproduces the zero-winding contributions to the BPS partition function of (1,1) Little String Theory wrapping an emergent torus. We therefore successfully test both the expressions for the $\mathcal{N}=1$ partition functions, as well as the relationship between the toroidal-quiver theory and Little String Theory through dimensional deconstruction.



# 1  Introduction

The application of the technique of supersymmetric localisation as a tool to compute partition functions for four-dimensional $\mathcal{N} = 2$ gauge theories on $S^4$ was a major technical advance with far-reaching consequences [1]. It has unlocked a whole new class of exact (all orders in the coupling) computations for a variety of gauge theories with extended supersymmetry, on a multitude of backgrounds and in diverse dimensions; for a review see [2]. However, it has been surprisingly difficult to apply this approach to minimally-supersymmetric theories on the four-sphere, which would be desirable if one were to extend its success to more realistic settings; see e.g. [3–5] for a discussion of associated issues.[1] Note that although $\mathcal{N} = 1$ partition functions on $S^4$ are known to be scheme dependent and therefore a priori ambiguous [8], one can still use them to extract meaningful physical information, cf. [3,9].

Given the above state of the art, it is then intriguing that the issues with the direct computation of the $\mathcal{N} = 1$ partition function on $S^4$ using localisation were recently sidestepped. In [10] known results for the exact, round-sphere partition functions of two- and three-dimensional theories with four supercharges were analytically continued to four dimensions. This culminated into specific expressions for the perturbative contribution of vector and chiral multiplets to the exact $S^4$-partition function for a particular class of $\mathcal{N} = 1$ theories. The proposal passes several checks, including a comparison with the holographic-dual results of [9] for the free energy of $\mathcal{N} = 1^*$ super Yang–Mills (sYM). Overall, this is an important first step in obtaining the full partition function of $\mathcal{N} = 1$ theories on $S^4$, although it seems difficult to extend the analytic-continuation argument to the nonperturbative piece; see [11–13] however, for alternative approaches in this direction.

In this note, we refine and test this proposal by making contact with dimensional deconstruction. We employ a "toroidal-quiver" theory with $\mathcal{N} = 1$ supersymmetry, obtained by orbifolding $\mathcal{N} = 4$ sYM by $\mathbb{Z}_{N_5} \times \mathbb{Z}_{N_6}$; this model fits within the calculational framework of [10]. In the deconstruction limit, this quiver theory is expected to give rise to the (1,1) Little String Theory (LST) on a torus [14]. Our goal is to provide a check of this conjecture at the level of exact partition functions.

We begin with the observation that, by writing the four-dimensional expressions of [10] in terms of multiple-gamma functions, these admit a natural refinement to the case of the ellipsoid, $S^4_{\epsilon_1, \epsilon_2}$. Schematically

$$\Gamma_3\left(x \middle| R^{-1}, R^{-1}, R^{-1}\right) \longmapsto \Gamma_3\left(x \middle| \epsilon_1, \epsilon_2, \tfrac{\epsilon_1 + \epsilon_2}{2}\right). \tag{1}$$

When the field content is such that one expects supersymmetry enhancement, this generalisation leads to the correct $\mathcal{N} = 2$ sYM partition functions on the ellipsoid, computed in [15]. We then provide a prescription for implementing the deconstruction limit directly on the perturbative partition function of the toroidal quiver, along the lines of [16]. The answer reproduces the zero-winding contributions to the Bogomol'nyi–Prasad–Sommerfeld (BPS) partition function of (1,1) LST on $S^4_{\epsilon_1, \epsilon_2} \times T^2$ and yields a first quantitative check for the deconstruction of LST from four-dimensional quiver-gauge theories.

The results of this note, summarised in Fig. 1, can be viewed as providing strong evidence for the validity of the analytic-continuation procedure of [10]. It would be very interesting to further test the applicability of our refined expressions to broader classes of $\mathcal{N} = 1$ theories, beyond those accessible through the calculational framework considered in that reference.

---

[1]It should be mentioned that such localisation calculations have been performed for four-dimensional theories with $\mathcal{N} = 1$ supersymmetry on other manifolds [6,7].

$$\mathcal{Z}^{\text{LST}}_{S^4_{\epsilon_1,\epsilon_2} \times T^2} \xleftarrow[\text{glueing}]{} \mathcal{Z}^{\text{LST}}_{\mathbb{R}^4_{\epsilon_1,\epsilon_2} \times T^2}$$

$$\Big\uparrow \text{deconstruction}$$

$$\mathcal{Z}^{\textbf{quiver}}_{S^4_{\boldsymbol{\epsilon_1,\epsilon_2}}} \xrightarrow[\text{enhancement}]{\text{SUSY}} \mathcal{Z}^{\mathcal{N}=2}_{S^4_{\epsilon_1,\epsilon_2}}$$

$$\Big\downarrow \text{unrefinement}$$

$$\mathcal{Z}^{\text{quiver}}_{S^4} \xleftarrow[\text{continuation}]{\text{analytic}} \mathcal{Z}^{\text{quiver}}_{S^D}$$

Figure 1: The $\mathcal{N}=1$ toroidal-quiver 4D partition function and its various limits.

## 2 $\mathcal{N}=1$ partition functions on the ellipsoid

The approach of [10] closely follows [1]. Starting with $\mathcal{N}=1$ sYM in ten-dimensional Minkowski space, one performs a dimensional reduction to Euclidean 2D/3D and reduces supersymmetry by explicitly performing two successive projections $\Gamma^{6789}\epsilon = \epsilon$, $\Gamma^{4589}\epsilon = \epsilon$, on the 32-component Killing spinors. The result is conformally mapped to the two/three-sphere by adding appropriate curvature couplings, leading to theories with one vector and three adjoint chiral multiplets; one can also turn on mass deformations in the latter.[2] Supersymmetric localisation yields the vector- and chiral-multiplet contributions on $S^2/S^3$; these can be combined to give the full partition function. The final two- and three-dimensional answer can be elegantly written in a D-dependent form, which can then be continued to D = 4. Through this operation one arrives at the proposed perturbative partition functions for four-dimensional theories with four supercharges [10].

The individual $\mathcal{N}=1$ four-dimensional vector- and chiral-multiplet contributions obtained via the above procedure[3] can be recast using multiple gamma functions, defined as the zeta-regularised product

$$\Gamma_N(x|\omega_1,\ldots,\omega_N) = \prod_{\boldsymbol{\ell}\in\mathbb{N}^N}^{\zeta} \frac{1}{x+\boldsymbol{\ell}\cdot\boldsymbol{\omega}} \tag{2}$$

for $\boldsymbol{\omega}\in\mathbb{R}^N_{>0}$, as

$$\mathcal{Z}^{\text{chi}}_{\text{pert}} = \prod_{\beta\in\mathcal{R}} \frac{\Gamma_3(R^{-1}+iM+i\langle\beta,\lambda\rangle|R^{-1},R^{-1},R^{-1})}{\Gamma_3(2R^{-1}-iM-i\langle\beta,\lambda\rangle|R^{-1},R^{-1},R^{-1})},$$

$$\mathcal{Z}^{\text{vec}}_{\text{pert}} = \prod_{\beta\in\text{Adj}} \frac{\Gamma_3(3R^{-1}-i\langle\beta,\lambda\rangle|R^{-1},R^{-1},R^{-1})}{\widehat{\Gamma}_3(i\langle\beta,\lambda\rangle|R^{-1},R^{-1},R^{-1})}, \tag{3}$$

where $\widehat{\Gamma}_N$ is a modification to (2) that only involves a product over $\boldsymbol{\ell}\in\mathbb{N}^N\setminus\{\mathbf{0}\}$. In the above, $R$ denotes the radius of the $S^4$, $M$ is a mass parameter, while $\lambda$ a variable with mass-dimension one that is an element of the Lie algebra to be integrated over in the final expression for the full partition function.[4] Finally, $\beta$ takes values in the weights of the representation $\mathcal{R}$ for the chiral multiplets, or the adjoint for the vector multiplet, and $\langle\beta,\lambda\rangle$ denotes an inner product in weight space.

---

[2]As viewed from ten dimensions, the directions $x^4,\ldots,x^9$ are transverse to the two/three-sphere.
[3]The relevant expressions can respectively be found in Eqs. (5.16) and (5.24) of [10].
[4]Our definition for the vector-multiplet partition function does not include the Haar measure.

The form of the partition functions in Eq. (3) suggests a natural refinement to the case of the ellipsoid, $S^4_{\epsilon_1,\epsilon_2}$:

$$\mathcal{Z}^{\text{chi}}_{\text{pert}} = \prod_{\beta \in \mathcal{R}} \frac{\Gamma_3(\epsilon_+ + iM + i\langle\beta,\lambda\rangle|\,\epsilon_1,\epsilon_2,\epsilon_+)}{\Gamma_3(2\epsilon_+ - iM - i\langle\beta,\lambda\rangle|\,\epsilon_1,\epsilon_2,\epsilon_+)} \, ,$$

$$\mathcal{Z}^{\text{vec}}_{\text{pert}} = \prod_{\beta \in \text{Adj}} \frac{\Gamma_3(3\epsilon_+ - i\langle\beta,\lambda\rangle|\epsilon_1,\epsilon_2,\epsilon_+)}{\widehat{\Gamma}_3(i\langle\beta,\lambda\rangle|\epsilon_1,\epsilon_2,\epsilon_+)} \, , \tag{4}$$

where $\epsilon_\pm = \frac{1}{2}(\epsilon_1 \pm \epsilon_2)$ and the ellipsoid $S^4_{\epsilon_1,\epsilon_2}$ is defined by

$$\frac{x_1^2}{R^2} + \frac{x_2^2 + x_3^2}{\epsilon_1^{-2}} + \frac{x_4^2 + x_5^2}{\epsilon_2^{-2}} = 1 \tag{5}$$

for $\epsilon_1, \epsilon_2 \in \mathbb{R}_{>0}$. Eqs. (4) readily reduce to the results for the round $S^4$, when $\epsilon_1 = \epsilon_2 = R^{-1}$.

Moreover, these refined expressions lead to the expected supersymmetry enhancement upon choosing a theory with a vector and a single massless adjoint chiral multiplet. Making use of the identity

$$\Gamma_2(x|\epsilon_1,\epsilon_2) = \frac{\Gamma_3(x|\epsilon_1,\epsilon_2,\epsilon_+)}{\Gamma_3(x + \epsilon_+|\epsilon_1,\epsilon_2,\epsilon_+)} \, , \tag{6}$$

one finds that

$$\mathcal{Z}^{\text{chi}}_{\text{pert}}|_{M=0} \; \mathcal{Z}^{\text{vec}}_{\text{pert}} = \prod_{\beta \in \text{Adj}} \Gamma_2(2\epsilon_+ - i\langle\beta,\lambda\rangle|\epsilon_1,\epsilon_2)^{-1} \, \widehat{\Gamma}_2(i\langle\beta,\lambda\rangle|\epsilon_1,\epsilon_2)^{-1} \, . \tag{7}$$

Once the Haar measure is also included this matches the answer for the $\mathcal{N} = 2$ vector-multiplet partition function on the ellipsoid, as calculated in [15].

Likewise, pairing up two chiral multiplets with the same mass in conjugate representations $\mathcal{R}$ results in

$$\mathcal{Z}^{\text{chi}}_{\text{pert}}|_{M,\lambda} \; \mathcal{Z}^{\text{chi}}_{\text{pert}}|_{-M,-\lambda} = \prod_{\beta \in \mathcal{R}} \Gamma_2(\epsilon_+ - iM - i\langle\beta,\lambda\rangle|\epsilon_1,\epsilon_2) \, \Gamma_2(\epsilon_+ + iM + i\langle\beta,\lambda\rangle|\epsilon_1,\epsilon_2) \, , \tag{8}$$

which matches the $\mathcal{N} = 2$ hypermultiplet result in [15].

Eqs. (4) form the seed for the perturbative partition function needed in dimensional deconstruction. The gauge theory of interest is obtained by twice-orbifolding the four-dimensional $\mathcal{N} = 4$ sYM with $U(KN_5N_6)$ gauge group by $\mathbb{Z}_{N_5} \times \mathbb{Z}_{N_6}$. The result is a toroidal quiver-gauge theory with $N_5 \times N_6$ $U(K)$ nodes and interconnecting bifundamental chiral multiplets as depicted in Fig. 2. It is at the "orbifold point", that is the coupling at each node takes the same value, which we will denote by $G$. It enjoys superconformal symmetry and is a Lagrangian example of the theories of class $\mathcal{S}_k$ [17, 18].

The dimensional-continuation argument can be directly applied to the $\mathcal{N} = 1$ toroidal-quiver theory: the two orbifold actions lead to the same Killing-spinor projections $\Gamma^{4589}\epsilon = \epsilon$, $\Gamma^{6789}\epsilon = \epsilon$,[5] and although they also break the gauge group to $U(K)^{N_5 \times N_6}$ with bifundamental chirals, these can be individually coupled to the curvature of $S^2/S^3$.

In accordance with the above, the perturbative part of the integrand of the full partition function for the toroidal-quiver theory on $S^4_{\epsilon_1,\epsilon_2}$ takes the form

$$\mathcal{Z}^{\text{quiv}}_{\text{pert}} = \prod_{b,\hat{b}} \Delta(\lambda^{(b,\hat{b})}) \, \mathcal{Z}^{(b,\hat{b})}_{\text{vec}} \, \mathcal{Z}^{(b,\hat{b})}_{\text{H}} \, \mathcal{Z}^{(b,\hat{b})}_{\text{D}} \, \mathcal{Z}^{(b,\hat{b})}_{\text{V}} \, , \tag{9}$$

---

[5]See [14] for the detailed action of the $\mathbb{Z}_{N_5} \times \mathbb{Z}_{N_6}$ orbifold.

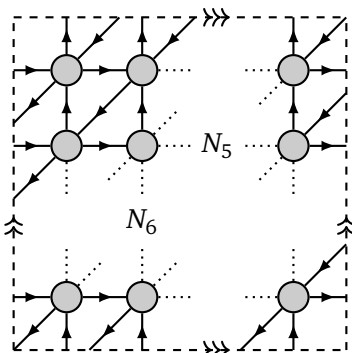

Figure 2: The quiver diagram for the $\mathbb{Z}_{N_5} \times \mathbb{Z}_{N_6}$ orbifold theory. Opposite edges are identified.

where

$$\Delta(\lambda^{(b,\hat{b})}) = \prod_{m \neq n} i(\lambda_m^{(b,\hat{b})} - \lambda_n^{(b,\hat{b})}) \tag{10}$$

is the Haar measure for each $U(K)_{(b,\hat{b})}$ gauge node, while the product in $b, \hat{b}$ runs over all $N_5 \times N_6$ nodes. The bifundamental chiral-multiplet contributions for the vertical (V), horizontal (H) and diagonal (D) terms can be organised according to their gauge-group representations, summarised in Tab. 1. The explicit expressions are given by[6]

$$\mathcal{Z}_{\mathrm{H}}^{(b,\hat{b})} = \prod_{m,n} \frac{\Gamma_3\big(\epsilon_+ + iM_{\mathrm{H}}^{(b,\hat{b})} + i\big(\lambda_m^{(b,\hat{b})} - \lambda_n^{(b+1,\hat{b})}\big)\big)}{\Gamma_3\big(2\epsilon_+ - iM_{\mathrm{H}}^{(b,\hat{b})} - i\big(\lambda_m^{(b,\hat{b})} - \lambda_n^{(b+1,\hat{b})}\big)\big)},$$

$$\mathcal{Z}_{\mathrm{D}}^{(b,\hat{b})} = \prod_{m,n} \frac{\Gamma_3\big(\epsilon_+ + iM_{\mathrm{D}}^{(b,\hat{b})} - i\big(\lambda_m^{(b,\hat{b})} - \lambda_n^{(b+1,\hat{b}+1)}\big)\big)}{\Gamma_3\big(2\epsilon_+ - iM_{\mathrm{D}}^{(b,\hat{b})} + i\big(\lambda_m^{(b,\hat{b})} - \lambda_n^{(b+1,\hat{b}+1)}\big)\big)},$$

$$\mathcal{Z}_{\mathrm{V}}^{(b,\hat{b})} = \prod_{m,n} \frac{\Gamma_3\big(\epsilon_+ + iM_{\mathrm{V}}^{(b,\hat{b})} + i\big(\lambda_m^{(b,\hat{b})} - \lambda_n^{(b,\hat{b}+1)}\big)\big)}{\Gamma_3\big(2\epsilon_+ - iM_{\mathrm{V}}^{(b,\hat{b})} - i\big(\lambda_m^{(b,\hat{b})} - \lambda_n^{(b,\hat{b}+1)}\big)\big)}, \tag{11}$$

where compared to (4) the weight-space inner products have been evaluated on the respective representations.[7] Similarly, the vector-multiplet expressions are given by

$$\mathcal{Z}_{\mathrm{vec}}^{(b,\hat{b})} = \prod_{m,n} \frac{\Gamma_3\big(3\epsilon_+ - i\big(\lambda_m^{(b,\hat{b})} - \lambda_n^{(b,\hat{b})}\big)\big)}{\widehat{\Gamma}_3\big(i\big(\lambda_m^{(b,\hat{b})} - \lambda_n^{(b,\hat{b})}\big)\big)}. \tag{12}$$

In the unrefined limit $\epsilon_1 = \epsilon_2 = R^{-1}$, and for $N_5 = N_6 = 1$, Eq. (9) collapses to the partition function of a single-node theory with a vector and three adjoint chiral multiplets with distinct

---

[6]From now on we will suppress the triple-gamma function parameters $\epsilon_1, \epsilon_2, \epsilon_+$ for brevity; the reader should assume that all expressions are given for the ellipsoid.

[7]We note that the lower-dimensional origin of the perturbative partition functions implies that the 4D mass parameters have to satisfy the constraints [10]

$$M_{\mathrm{H}}^{(b,\hat{b})} + M_{\mathrm{D}}^{(b,\hat{b})} + M_{\mathrm{V}}^{(b+1,\hat{b})} = 0,$$

$$M_{\mathrm{H}}^{(b,\hat{b}+1)} + M_{\mathrm{D}}^{(b,\hat{b})} + M_{\mathrm{V}}^{(b,\hat{b})} = 0.$$

Our deconstruction prescription will respect these conditions.

Table 1: The chiral multiplets for the toroidal quiver and their gauge-group representations.

|  | $U(K)_{(b,\hat{b})}$ | $U(K)_{(b,\hat{b}+1)}$ | $U(K)_{(b+1,\hat{b}+1)}$ |
|---|---|---|---|
| $V^{(b,\hat{b})}$ | $\square$ | $\overline{\square}$ | $\mathbf{1}$ |
| $H^{(b,\hat{b}+1)}$ | $\mathbf{1}$ | $\square$ | $\overline{\square}$ |
| $D^{(b,\hat{b})}$ | $\overline{\square}$ | $\mathbf{1}$ | $\square$ |

masses, i.e. $\mathcal{N}=1^*$ sYM as in [10]. In a different limiting case where $N_5=1$ or $N_6=1$, and after appropriately tuning the masses, the theory reduces to an $\mathcal{N}=2$ circular quiver involving the expected partition function Eqs. (7), (8).

## 3 Deconstruction and Little String Theory

We next set up a nontrivial test of Eq. (9) using dimensional deconstruction; the procedure of creating geometric extra dimensions from closed quiver-gauge theories at long distances, along the Higgs branch [19].[8] Following [14], we would like to explore how the four-dimensional $\mathcal{N}=1$ toroidal-quiver gauge theory deconstructs the six-dimensional (1,1) LST and test their proposal by comparing the respective partition functions.

Dimensional deconstruction dictates that the horizontal and vertical chiral multiplets in Fig. 2 acquire vacuum expectation values $v_5$ and $v_6$, and that one takes the limit

$$N_5, N_6 \to \infty , \qquad v_5, v_6 \to \infty , \qquad G \to \infty , \tag{13}$$

while fixing the ratios

$$2\pi R_5 \equiv \frac{N_5}{Gv_5} , \qquad 2\pi R_6 \equiv \frac{N_6}{Gv_6} . \tag{14}$$

The latter are identified with the radii of the deconstructed compact dimensions. The theory that emerges at low energies is six-dimensional sYM with (1,1) supersymmetry—the amount of supersymmetry has quadrupled—and gauge coupling $g_6^2 = (v_5 v_6)^{-1}$. However, on top of the conventional spectrum of massive gauge bosons that deconstructs the Kaluza–Klein (KK) towers associated with the square torus $S_{R_5}^1 \times S_{R_6}^1$, the four-dimensional quiver contains additional towers of states. This can be seen by acting on the KK towers with the S-duality transformation of the four-dimensional toroidal-quiver theory—sending $G \mapsto N_5 N_6/G$. From the point of view of the six-dimensional theory, this is a T-duality transformation and the new towers are interpreted as string winding modes on the torus. The combined spectrum of this non-gravitational theory matches that of (1,1) LST with string tension $1/\alpha' = 1/g_6^2$. This remarkable conclusion can also be reached using a complementary brane-engineering argument [14] but to our knowledge there exist no quantitative checks of this claim to date. The partition function Eq. (9) immediately allows for such a possibility, as we will now show.

A prescription for implementing the deconstruction limit at the level of four-dimensional partition functions was given in [16]. Applied to (9), it involves identifying the parameters under the products (for $b \le c$, $\hat{b} \le \hat{c}$) as

$$\lambda_m^{(b,\hat{b})} - \lambda_n^{(c,\hat{c})} = \lambda_m - \lambda_n , \qquad M_V^{(b,\hat{b})} = 0 , \qquad M_H^{(b,\hat{b})} = -M_D^{(b,\hat{b})} = M , \tag{15}$$

---

[8]Recent work that uses deconstruction to study connections between theories across dimensions includes [20, 21].

along with the following shift in the argument for each triple-gamma function

$$\Gamma_3(x^{(b,\hat{b})}) \longmapsto \Gamma_3(x^{(b,\hat{b})} + 2\pi i b R_5^{-1} + 2\pi \hat{b} R_6^{-1}) \,. \tag{16}$$

Furthermore, one needs to extend the range of the products over $b, \hat{b} \in \mathbb{Z}$ when taking the limit $N_5, N_6 \to \infty$.

We will explicitly perform the resultant product over $b, \hat{b}$ by zeta-function regularisation. In particular, we will use [22]

$$\prod_{b,\hat{b} \in \mathbb{Z}}^{\zeta} \Gamma_N(x + b + \hat{b}\tau | \omega_1, \ldots, \omega_N) = \prod_{\ell \in \mathbb{N}^N} \frac{1}{\theta(q|y)} \,, \tag{17}$$

with

$$y = e^{-2\pi i(x+\ell\cdot\boldsymbol{\omega})} \,, \qquad q = e^{2\pi i\tau} \tag{18}$$

and $\theta(q|y)$ a $q$-theta function. Applying (17) to the triple-gamma functions that appear in (9), and after some algebra, the result for the integrand of the perturbative partition function of the toroidal theory in the deconstruction limit becomes

$$\mathcal{Z}_{\text{pert}}^{\text{dec}} = \left( \frac{\prod\limits_{\ell \in \mathbb{N}^2} \theta\big(q\big|\mathfrak{q}^{\ell_1+1}\mathfrak{t}^{\ell_2+1}\big) \prod\limits_{\ell \in \mathbb{N}^2\backslash\{\mathbf{0}\}} \theta\big(q\big|\mathfrak{q}^{\ell_1}\mathfrak{t}^{\ell_2}\big)}{\prod\limits_{\ell \in \mathbb{N}^2} \theta\big(q\big|\mathfrak{q}^{\ell_1+\frac{1}{2}}\mathfrak{t}^{\ell_2+\frac{1}{2}}Q_M\big) \, \theta\big(q\big|\mathfrak{q}^{\ell_1+\frac{1}{2}}\mathfrak{t}^{\ell_2+\frac{1}{2}}Q_M^{-1}\big)} \right)^K$$

$$\times \prod_{m\neq n} \frac{\prod\limits_{\ell \in \mathbb{N}^2} \theta\big(q\big|\mathfrak{q}^{\ell_1+1}\mathfrak{t}^{\ell_2+1}\xi_{mn}\big) \, \theta\big(q\big|\mathfrak{q}^{\ell_1}\mathfrak{t}^{\ell_2}\xi_{mn}\big)}{\prod\limits_{\ell \in \mathbb{N}^2} \theta\big(q\big|\mathfrak{q}^{\ell_1+\frac{1}{2}}\mathfrak{t}^{\ell_2+\frac{1}{2}}Q_M\xi_{mn}\big) \, \theta\big(q\big|\mathfrak{q}^{\ell_1+\frac{1}{2}}\mathfrak{t}^{\ell_2+\frac{1}{2}}Q_M^{-1}\xi_{mn}\big)}$$

$$\tag{19}$$

where $\tau = iR_5/R_6$ and with the following definitions for the fugacities:

$$\mathfrak{q} = e^{-R_5\epsilon_1} \,, \qquad\qquad \mathfrak{t} = e^{-R_5\epsilon_2} \,,$$
$$Q_M = e^{iR_5 M} \,, \qquad\qquad \xi_{mn} = e^{iR_5(\lambda_m - \lambda_n)} \,. \tag{20}$$

This concludes the application of the deconstruction prescription to the $\mathcal{N} = 1$ quiver partition function on $S^4_{\epsilon_1,\epsilon_2}$.

## 4 The BPS partition function for (1,1) Little String Theory

In the final part of this note we will show how Eq. (19) compares against (1,1) LST, the BPS partition function for which can be calculated by appealing to topological string theory: via a chain of dualities, the topological string partition function for the toric Calabi–Yau threefold depicted by the dual toric diagram in Fig. 3 evaluates the BPS partition function of (1,1) LST on $\mathbb{R}^4_{\epsilon_1,\epsilon_2} \times T^2$; cf. [23–25].[9]

A lengthy calculation using the refined topological vertex formalism [26] leads to a result that factorises into winding sectors, accounting for little strings that wrap one of the toroidal directions. We find

$$\mathcal{Z}^{\text{LST}}_{\mathbb{R}^4_{\epsilon_1,\epsilon_2} \times T^2} = \widehat{\mathcal{Z}} \mathcal{Z}^{\text{wind}} \,, \tag{21}$$

---

[9]The web diagram for $K$ NS5 branes in the Coulomb branch coincides with the dual toric diagram of Fig. 3, where the vertical directions correspond to NS5 branes.

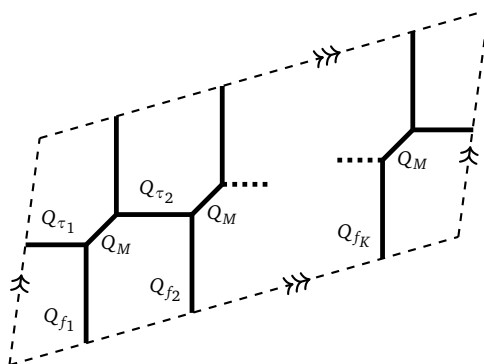

Figure 3: The dual toric diagram for the rank-$K$ LST. The vertical direction has been chosen as the preferred one. There are no non-trivial framing factors. Each internal line is associated with a Kähler parameter and a partition. The periodic identification of both horizontal and vertical lines is indicated by the cyclic identification of Kähler parameters and their associated partitions.

where[10]

$$\widehat{\mathcal{Z}} = \left( Q_M \sqrt{\frac{\mathfrak{t}}{\mathfrak{q}}} \right)^{-\frac{K}{24}} \text{P.E.} \left[ \frac{q}{1-q} + \frac{I_+}{2} \frac{1+q}{1-q} \sum_{m,n=1}^{K} \xi_{mn} \right], \tag{22}$$

with

$$I_+ = \frac{\mathfrak{q} + \mathfrak{t} - \sqrt{\mathfrak{q}\mathfrak{t}}(Q_M + Q_M^{-1})}{(1-\mathfrak{t})(1-\mathfrak{q})} \tag{23}$$

and

$$\mathcal{Z}^{\text{wind}} = \sum_{k=0}^{\infty} w^k \mathcal{Z}^{(k)}, \tag{24}$$

such that

$$\mathcal{Z}^{(k)} = \sum_{Y: \sum_m |Y_m|=k} \prod_{m,n=1}^{K} \prod_{s \in Y_m} \frac{\theta_1\left(\tau|E_{mn}(s)+M-\epsilon_-\right) \theta_1\left(\tau|E_{mn}(s)-M-\epsilon_-\right)}{\theta_1\left(\tau|E_{mn}(s)+\epsilon_2\right) \theta_1\left(\tau|E_{mn}(s)-\epsilon_1\right)}. \tag{25}$$

In the last expression one has that $\mathcal{Z}^{(0)} = 1$ and

$$E_{mn}(s) = \lambda_m - \lambda_n - \epsilon_1 h_m(s) + \epsilon_2 v_n(s), \tag{26}$$

with $s$ denoting a box in the coloured Young diagram $Y_m$ that labels a given partition, $h_m(s)$ the horizontal distance from the box $s$ to the right edge of $Y_m$ and $v_n(s)$ the vertical distance to the bottom edge.

To reach this answer using the topological-vertex formalism one also needs to make the following identifications between the parameters appearing in the toric diagram and the physical ones:

$$Q_M Q_{\tau_{\alpha>1}} = e^{iR_5(\lambda_{\alpha-1}-\lambda_\alpha)}, \qquad \prod_{i=1}^{K} Q_{\tau_i} Q_M = q,$$

$$Q_M Q_{\tau_1} = q e^{iR_5(\lambda_K-\lambda_1)}, \qquad \prod_{\alpha=1}^{K} (Q_{f_\alpha} Q_M)^{|Y_\alpha|} = w^k, \tag{27}$$

---

[10]The Plethystic Exponential is defined as the following operation P.E.$[f(t)] = \exp\left( \sum_{n=1}^{\infty} \frac{1}{n} f(t^n) \right)$.

while keeping in mind that

$$\mathfrak{q} = e^{-R_5\epsilon_1}, \qquad \mathfrak{t} = e^{R_5\epsilon_2}, \tag{28}$$

for $\epsilon_1 \in \mathbb{R}_{>0}$ and $\epsilon_2 \in \mathbb{R}_{<0}$.

Two copies of the BPS partition functions of LST on $\mathbb{R}^4_{\epsilon_1,\epsilon_2} \times T^2$ can be "glued together" to construct the corresponding expressions on the ellipsoid [1, 27, 28]:

$$Z_{S^4_{\epsilon_1,\epsilon_2}} = \int [d\lambda] \, \mathcal{Z}_{\mathbb{R}^4_{\epsilon_1,\epsilon_2}} \, \overline{\mathcal{Z}_{\mathbb{R}^4_{\epsilon_1,\epsilon_2}}}, \tag{29}$$

with

$$\overline{\mathcal{Z}_{\mathbb{R}^4_{\epsilon_1,\epsilon_2}}}(q, \mathfrak{t}, \mathfrak{q}, Q_M, \lambda_m) = \mathcal{Z}_{\mathbb{R}^4_{\epsilon_1,\epsilon_2}}(q, \mathfrak{t}^{-1}, \mathfrak{q}^{-1}, Q_M^{-1}, -\lambda_m), \tag{30}$$

for $\mathrm{Re}\,\tau = 0$.

Now recall that at low energies (1,1) LST reduces to (1,1) sYM in six dimensions. In turn, the BPS partition function for LST coincides with that of (1,1) sYM, with the zero-winding contributions in the former being reproduced by the perturbative piece in the latter, while the nontrivial winding sectors coming from the tower of instanton-string states, see e.g. [24]. Therefore, for the purpose of comparing with the deconstruction result we will single out the zero-winding sector of the LST partition function.

Since the glueing prescription Eq. (29) can be implemented independently for each winding sector, it is straightforward to extract the zero-winding piece of $\mathcal{Z}^{\mathrm{LST}}_{S^4_{\epsilon_1,\epsilon_2} \times T^2}$ from (21). All in all it can be shown that the 0-winding contributions to the integrand of Eq. (29) is given by

$$\mathcal{Z}^{\mathrm{LST, \, 0\text{-}wind}}_{S^4_{\epsilon_1,\epsilon_2} \times T^2} = \frac{e^{-\frac{\pi i \tau}{6}}}{\eta(\tau)^2} \prod_{m,n=1}^{K} \prod_{\ell \in \mathbb{N}^2} \frac{\theta\big(q\big|\mathfrak{q}^{\ell_1+\frac{1}{2}}\mathfrak{t}^{\ell_2+\frac{1}{2}}Q_M\xi_{mn}\big) \, \theta\big(q\big|\mathfrak{q}^{\ell_1+\frac{1}{2}}\mathfrak{t}^{\ell_2+\frac{1}{2}}Q_M^{-1}\xi_{mn}\big)}{\theta\big(q\big|\mathfrak{q}^{\ell_1}\mathfrak{t}^{\ell_2+1}\xi_{mn}\big) \, \theta\big(q\big|\mathfrak{q}^{\ell_1+1}\mathfrak{t}^{\ell_2}\xi_{mn}\big)}. \tag{31}$$

As a last step, one needs to analytically continue the above to $\epsilon_2 \in \mathbb{R}_{>0}$ before comparing with Eq. (19). Proceeding as e.g. in App. A.2 of [29], the LST result Eq. (31) coincides with the one from deconstruction up to an overall geometric factor of $\exp(-\pi i \tau/6)\,\eta(\tau)^{-2}$.[11]

# Acknowledgements

We would like to thank Jan Peter Carstensen, Vasilis Niarchos and Elli Pomoni for useful discussions and collaboration on related topics. J.H. is supported by an STFC research studentship and would like to thank the CERN Theory Group for hospitality during the last stages of this work. R.P. is supported by a Queen Mary College research studentship. C.P. is supported by the Royal Society through a University Research Fellowship.

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
