# Peer review of "Deconstructing Little Strings with $\mathcal{N}=1$ Gauge Theories on Ellipsoids"

_SciPost Physics, doi:SciPost Phys. 4, 042 (2018)_

## Round 2 · Referee Report · Brandon Robinson · 2018-5-22

Strengths
1 - The results presented are novel and impactful in an established and active field of research using a variety of techniques to accomplish their construction.
2- The work required to establish the framework for the test of proposed relationships for $\mathcal{N}=1$ partition functions of gauge theories on curved backgrounds is no small feat and should celebrated.
3- Incidentally through building the toroidal quiver gauge theory partition function and following the deconstruction procedure, the authors propose through their results a quantitative check of a brane construction of the six-dimensional theory, which is noteworthy.
4 - The arguments being made are logically consistent and presented in a way that makes, overall, the work compelling to read.
5 - The cited references are appropriate and provide a sufficient background for further questions that a reader may be inspired to take up.
Weaknesses
1 - The compact writing style has left much of the detail of the calculations and their subtleties unstated.
2 - Much of the reference to work not done by the authors is left unqualified, vague, and without further guide for the reader.
3 - There does not seem to be much advantage to adopting two-column style, and given the length of some of the equations and on-line expressions, it seems unnecessarily cumbersome.
4 - Some of the figures and tables seem unnecessary and could be replaced by clearer explanation within the text - Table 1 and Figure 2 in particular.
Report
The work in ''Deconstructing Little Strings with $\mathcal{N}=1$ Gauge Theories on Ellipsoids'' is, in itself, laudable. The authors' work to perform non-trivial checks of an interesting proposal in which the perturbative part of a certain topological sector of an $\mathcal{N}=1$ gauge theory on a curved manifold can be analytically continued to higher dimensions from a base expression in two-dimensions. The authors
There are, however, a few points with regards to the discussion of the toroidal quiver and the node-wise use of supersymmetric localization to build the deconstructed description that should be addressed. First, there are minor points that do not effect the end result, but should be attended to increase the precision of the language in the text:
1 - In the first paragraph of the introduction, ''[Supersymmetric localisation] has unlocked a whole new class of exact (all order in the coupling) [...]'' is a bit misleading. As a general conceit- including the calculation of the equivariant Euler class of the normal bundle to the locus - supersymmetric localization does not need to reference any perturbative expansion but rather produces a result for the [1-loop] exact partition function even in strong coupling regimes where ''all orders'' is not a useful concept - see e.g. applications of supersymmetric localization in the context of holography. Although the above comment is understandable that for the purposes of the current work the view will always be on perturbative aspects of the localized theories, it misses some of the power of supersymmetric localization.
2 - In the first paragraph of the section titled ''$\mathcal{N} = 1$ Partition Functions on the Ellipsoid'', it is stated that ''Supersymmetric localisation yields the vector - and chiral-multiplet contributions [...]''. The quoted statement is again a bit misleading because the contributions that are listed in the following paragraph in eq. (3) are the contributions of the vector- and chiral-multiplets to the 1-loop fluctuations around the theory that results from the localization procedure. If the quoted statement is meant to be understood as synecdoche, eq. (3) should be clearly noted as the ''1-loop'' contribution.
3 - In the explanation below eq. (3), there is a statement that ''$\lambda$'' is an element of the Lie Algebra that will be integrated over in the final expression for the full partition function. While in eq. (29) the integration of $\lambda$ is explicit, the partition functions in eqs. (31) and (19) still appear to retain some $\lambda$ dependence through $\xi_{mn} = \lambda_m - \lambda_n$. This causes a tension that does not seem to be resolved in the text.
4- Also, in eq. (31) there is a contribution that appears with no explication: $\eta(\tau)^{-2}$. This also appears as a point of tension in that there is an overall multiplicative factor that is said to be ''dropped in the topological string literature'', but if one were to want to claim an exact match of the partition functions in eqs. (19) and (31), there needs to be better justification for dropping the overall $e^{-\frac{\pi i \tau}{6}}\eta(\tau)^{-2}$.
5- In the paragraph below eq. (12), the paragraph presents a sort of tautology in that since the toroidal quiver theory is constructed from chaining together refined, single-node results with appropriate bifundamental chiral multiplet contributions, it must reproduce the result of [10] when limiting to the single, unrefined node.
6 - There is an ambiguity leading up to the result in eq. (19), where, in the paragraph starting with ''Dimensional deconstruction dictates that [...]'' on page 4, a sentence starts '' The combined spectrum of this non-gravitational theory [...]''. However, the discussion the precedes the reference to the ''combined spectrum'' surrounds the six-dimensional $\mathcal{N}=(1,1)$ super-Yang-Mills theory (SYM), the Kaluza-Klein (KK) modes, and - immediately before the sentence in question - a discussion about the S-dual spectrum. It appears that the ''combined spectrum'' is pointing to the SYM + KK + S-dual spectra, but should only be referring to SYM + KK.
7 - In the sentence immediately after eq. (16), the extension of $(b,\,\hat{b})\in \mathbb{Z}$ na\"ively looks to be double counting modes on the torus, but there is no qualification of why this is prescriptively necessary.
8 - One slightly nagging point is that while what appear to be the 1-loop contributions around the localized theory are displayed in eqs. (3), there is no mention of what fields are specifying the locus nor what the ''classical'' theory at the locus actually is.
9 - In the introduction it is said in the first paragraph that ''meaningful information'' can be extracted from the scheme-dependent partition function. The lede is buried here in that, the unambiguous comparison for localized gauge theories on an $S^4$ is the third derivative of the free energy with respect to a relevant energy scale.
Beyond those minor points, there is an oversight in the discussion that begins "Eqs. (4) form the seed [...]'' at the top of page 3 that must be addressed. In taking the deconstruction limit of the toroidal quiver theory, it is by its nature ensuring that parent $U(K N_5 N_6)$ gauge group is in the large rank limit. However on the quiver, it is not necessary that the $U(K)$ theory on each node is in the large $K$ limit. This leaves the text as it stands with a bit of a puzzle. Either the large $K$ limit is assumed to hold on every node on the toroidal quiver, or there is a claim - hidden in the work - about cancellations of the instanton contributions amongst the nodes over the entirety of the quiver. The latter possibility seems implausible and would require significant work to justify. The former possibility - assuming large $K$ - still must be mentioned as it is crucial in sufficiently suppressing instanton contributions to the localized theory.
Pending the corrections and clarifications requested being made - some of which are strictly optional - I would endorse the publication of the work in this journal.
Requested changes
1 - Address the issues regarding the physics and language used to describe it raised in the report above.
2 - Improve the transparency of what ''expressions'' are required from the cited work in order to perform the calculations. This could be something as little as directing the reader to the section or equation within the cited work where appropriate. The place where this is most egregious is with regards to the most frequently cited work, [10], which is lengthy and not simple to parse in terms of what is necessary for the present work.
3 - Optionally, for the notably complicated or subtle calculations, e.g. the refined topological vertex formalism for the BPS partition function of the $(1,1)$ Little String Theory, an appendix would be useful so that the ''lengthy calculation'' feels less like a black box. While this suggestion is not a strictly necessary change, would make the work more self-contained.
4 - For the purposes of readability, the appearance of a new symbol or expression should have its role explained in a proximate section of text. For example, the symbol $\tau$ appears in eq. (17), but it is not defined until after eq. (19) halfway down the page (and in another column). Another example is the discussion of the projections $\Gamma^{6789}\epsilon = \epsilon$ and $\Gamma^{4589}\epsilon = \epsilon$ in the first paragraph of page 2: no conventions for the ten-dimensional geometry have been given at this point, and only later in eq. (5) is any relevant discussion of the four-dimensional geometry given. While footnote 2 does give some convention, it should be moved to the end of the previous sentence.
5 - Optionally, Fig. 2 or Table 1 should be removed and the discussion their respective contents should be made clearer in the text. Table 1 seems to be the most redundant in that by more clearly labeling the links of the quiver in Fig. 2, the ''bifundamental'' nature of the links should be obvious.
6 - Optionally given the compact nature of the text, in lieu of a conclusion/summary section, if the authors have any thoughts about open questions or where they see the program being put to further use, it would provide a nice coda to the work.

---

## Round 2 · Referee Report · Anonymous · 2018-5-30

Strengths
1- Clarity of the discussion and computations
2- Short paper
Weaknesses
1- The result is nice but not that impressive.
Report
The paper proposes a test of the deconstruction "duality" relating the 6d (1,1) Little String Theory (LST) on a two-torus to a certain 4d $\mathcal{N}=1$ toroidal quiver theory on its Higgs branch in a limit of infinite number of gauge nodes.
The test is performed by comparing the $S^4$ and $S^4\times T^2$ partition functions in the deconstruction limit.
Because supersymmetric localization techniques are not available for $\mathcal{N}=1$ theories on $S^4$, the authors rely on recent results using analytical continuation in spacetime dimension to extract the perturbative part of (the integrand of) $Z(S^4)$, which they generalize to squashed $S^4$.
This is compared to the zero winding sector of LST on $S^4\times T^2$, which is taken from known topological string results.
The computation is presented in a concise way, but with all the important steps clearly stated. The authors find a non-trivial agreement, which supports the deconstruction proposal.
The last equation (31) still depends on the matrix model integration variables $\lambda_n$. I assume this is the integrand of the $S^4 \times T^2$ partition function (which is matched with the integrand of the $S^4$ partition function).
One comment is that on the 4d side the authors make use of analytical continuation in dimension but the result used is only the one-loop determinant on $S^4$, which most likely can be directly computed. Similarly on the 6d side, the zero winding sector is presumably the one-loop determinant of 6d $\mathcal{N}=(1,1)$ SYM on $S^4\times T^2$, which might not necessitate topological strings.
The match remains non-trivial and constitutes a first important step towards matching the full partition functions.
Requested changes
1- Clarification on the integrand vs integral issue in equation (31).

---

## Round 2 · Referee Report · Anonymous · 2018-6-4

Strengths
-Probes localization for N=1 theories by studying an $\mathcal{N}=1$ theory deconstructing LST
-Finds nice agreement with results from the topological vertex, thus supporting the $\mathcal{N}=1$ proposed formulas as well as the deconstruction of LST
Weaknesses
- No real weakness
Report
The paper is well-written, interesting and concise. It offers a nice test to the $\mathcal{N}=1$ formulas of ref.[10] and at the same time probes deconstruction of LST. It is true that the N=1 studied theory is very particular: an orbifold of $\mathcal{N}=4$. For instance, even though $\mathcal{N}=1$, fields develop no large anomalous dimensions. Hence, it would be interesting to study it for more general theories (and in particular compare with the holographic result). Also, the authors study the limit in which the quiver is large in both dimensions of the torus ($N_5$, $N_6$ both go to infinity). It would be interesting to study the limit of, say, $N_5$ to infinity while $N_6$ constant, presumably deconstructing an orbifold of the (2,0) theory.
Requested changes
- No changes requested

---

## Round 3 · Author Response

Dear Editor,

We thank all referees for a careful reading of the manuscript along with the associated comments and suggestions. We have implemented a list of changes to our submission based on these.

Sincerely,

J. Hayling, R. Panerai and C. Papageorgakis

---

## Round 3 · List of Changes

We order our response by referee report:

Referee 1:

We first reply to the points raised in the main report below:

1 - We believe that the phrasing in this instance is sufficiently clear; the qualifier “exact (to all orders in the coupling)” refers to the localisation of supersymmetric gauge theories where such a concept exists (and not to the localisation technique in full generality). We have not implemented any changes associated with this point.

2 - We feel that the phrasing “1-loop contribution” is misleading because: 1) it would imply that this calculation is the result of bona fide localisation calculation in 4D (see also point 8 below) and 2) even in that event the 1-loop contributions refer to the functional determinants of the free limit for the deformed theory. The qualifier “vector- and chiral-multiplet contributions” is more appropriate, in our opinion, as it captures the full contributions to the end result from these multiplets in the zero-instanton sector. We have not implemented any changes associated with this point.

3 - We thank the referee for raising this point, as the statement in the paper was misleading. We have stressed both above Eq. (9) as well as above Eq. (19) that the quantities presented are for the integrand of the perturbative partition function. There was also a notation clash towards the end of the letter, where the integrated LST partition function Eq. (29) and the integrand Eq. (31) were denoted with the same symbol. We have changed the former to resolve this ambiguity and stressed that Eq. (31) is to be integrated over $\lambda$.

4- We believe that it is clearly stated in the text that the two results match up to this overall factor. This is a purely geometric contribution that is not dropped for the sole purpose of ensuring the matching of the two calculations presented in our work, but often dropped in the topological strings literature as it contains no dynamical information. We have clarified this point in Footnote 11.

5- This paragraph was not meant to demonstrate tests of our proposal, but illustrate how this is a generalisation of known results. We have changed the phrasing slightly to clarify this.

6 - This sentence is indeed referring to SYM + KK + S-dual spectra. The S-dual spectra are needed in order to reproduce the winding sectors of LST as in [14], the deconstruction of which we are reviewing at this stage without restricting to the 0-winding sector. We have not implemented any changes associated with this point.

7 - We interpret this comment to refer to $b$ and $\hat b$ taking values in $\mathbb Z$ instead of $\mathbb N$, in which case we are not double counting as this is reproducing all the KK modes of the higher-dimensional theory (positive and negative), as e.g. in the original paper of Arkani-Hamed, Cohen and Georgi below equation (2.8). We have not implemented any changes associated with this point.

8 - There is no known way to localise these $\mathcal N=1$ quiver theories on $S^4$, and therefore no associated 4D localisation locus. The integration variables are inherited from the 2D/3D calculation of [10], where such a locus exists, via analytic continuation. This is also why we chose not to use the 1-loop nomenclature in point 2 above. We have not implemented any changes associated with this point.

9 - The number of derivatives that needs to be taken depends on the quantity that is probed, as can be easily found in our references [3] and [9]. We find that providing all these details in the opening paragraph of this letter would detract from introducing the topic to a general audience. The interested reader can find more information in the references provided. We have not implemented any changes associated with this point.

For the final point in this section of the report:

There is no need to take the large-rank limit on each node for our arguments to go through (large K). Our claim is not that the instantons are suppressed but that we are only reproducing the zero-winding sector of LST. Therefore the non-zero-winding sectors are very much there but we do not have a four-dimensional derivation for them. We have not implemented any changes associated with this point.

Moving on to the Requested Changes section:

1 - Addressed as above.

2 - We have included a new footnote 3 addressing this point.

3 - Since this is a letter submission, we decided not to include this calculation as it would significantly increase the length of the paper and they can be reproduced using standard technology from the literature. We have not implemented any changes associated with this point.

4 - Eq. (17) presents a mathematical identity that holds for any $\tau$, provided that $\Im \tau>0$. We specify $\tau$ only when we apply this to the case in hand, namely from Eq. (19) onwards. We have not implemented any changes associated with this point. However, we have clarified the ten-dimensional geometry at the beginning of that paragraph and the notation used for the susy projections should now be evident. This also follows the conventions of [10].

We hope that we have sufficiently addressed the referee’s concerns.

Referee 2:

This point was raised and addressed in the response to the first referee (point 3; main report). We include it here once again for ease of navigation:

We thank the referee for raising this point, as the statement in the paper was misleading. We have stressed both above Eq. (9) as well as above Eq. (19) that the quantities presented are for the integrand of the perturbative partition function. There was also a notation clash towards the end of the letter, where the integrated LST partition function Eq. (29) and the integrand Eq. (31) were denoted with the same symbol. We have changed the former to resolve this ambiguity and stressed that Eq. (31) is to be integrated over $\lambda$.

Referee 3:

No changes requested.

---

## Editorial Decision

published